# Morphological and Functional Characteristics of the Peroneus Muscles in Patients with Lateral Ankle Sprain: An Ultrasound-Based Study

**DOI:** 10.3390/medicina58010070

**Published:** 2022-01-03

**Authors:** Satoshi Arima, Noriaki Maeda, Makoto Komiya, Tsubasa Tashiro, Kazuki Fukui, Kazuki Kaneda, Mitsuhiro Yoshimi, Yukio Urabe

**Affiliations:** Department of Sports Rehabilitation, Graduate School of Biomedical and Health Sciences, Hiroshima University, 1-2-3 Kasumi, Minami-ku, Hiroshima 734-8553, Japan; satoshi-arima4646@hiroshima-u.ac.jp (S.A.); norimmi@hiroshima-u.ac.jp (N.M.); makoto-komiya@hiroshima-u.ac.jp (M.K.); tsubasatashiro716@hiroshima-u.ac.jp (T.T.); kazuki-fukui@hiroshima-u.ac.jp (K.F.); kazuki-kaneda@hiroshima-u.ac.jp (K.K.); mitsuhiroyoshimi0116@hiroshima-u.ac.jp (M.Y.)

**Keywords:** ankle evertor strength, lateral ankle sprain, peroneus brevis, peroneus longus, peroneus muscles, sports injury, ultrasound

## Abstract

*Background and Objectives*: The effectiveness of multiple ultrasound evaluations of the peroneus muscles morphology, including muscle cross-sectional area (CSA) and connective tissue, after lateral ankle sprain (LAS) is unknown. This study aimed to measure the peroneus muscles after LAS at three points, adding distal 75% to the conventional measurement points, in order to obtain a detailed understanding of the post-injury morphology and to propose a new evaluation index of the peroneus muscles for multiple LAS. *Materials and Methods*: Participants with and without LAS (LAS and control groups, 16 each) were recruited. The muscle cross-sectional area (CSA) and muscle echogenicity were measured using a B-mode ultrasound system at 25%, 50%, and 75% proximal to the line connecting the fibular head to the lateral malleolus. The ankle evertor strength was measured using a handheld dynamometer. Simultaneously, the peroneus longus (PL) and peroneus brevis (PB) muscle activities were measured using surface electromyography. Measurements for the LAS side, non-LAS side, and control leg were performed separately. *Results*: The CSA was significantly higher at 75% on the LAS side than on the non-LAS side and in the control leg. Muscle echogenicity of the LAS side at 75% was significantly lower than that of the non-LAS side and the control leg. Muscle activity of the PL was significantly lower and the PB was higher on the LAS side than on the non-LAS side and in the control leg. *Conclusions*: The PL was less active than the PB, while the PB was found to be overactive, suggesting that PB hypertrophy occurs due to an increase in the percentage of muscle fibers and a decrease in the connective tissue. Therefore, it is necessary to evaluate the condition of the PL and PB separately after LAS.

## 1. Introduction

Lateral ankle sprain (LAS) is one of the most frequent sports-related musculoskeletal injuries [1]. In a 16-year study of United States college athletes, the incidence of LAS was 0.83/1000 athlete exposures, accounting for 15% of all injuries. In fact, LAS is the type of injury with the highest incidence [2]. In a study that investigated symptoms 7 years after LAS, 32% of participants complained of chronic pain, swelling, or recurrent LAS, and 72% of these participants had limited their sports activities [3]. In addition, LAS is a serious musculoskeletal injury because of its high recurrence rate of 73.5% [4], which may leave post-injury sequelae and affect sports activities. It has also been reported that patients with LAS have differences in the functional characteristics of muscles around the ankle such as the peroneus muscles and tibialis anterior compared to healthy people [5]. Therefore, in order to prevent recurrence of LAS, it is necessary to accurately assess the status of the muscles around the ankle after LAS and then provide appropriate treatment.

The peroneus muscles, such as the peroneus longus (PL) and peroneus brevis (PB), are, in particular, the focus of post-LAS evaluation because of their ability to counteract ankle eversion. Although methods such as the muscle activity assessment [6], ankle evertor strength assessment [7], and peroneus muscles reaction time measurement [8] have been reported to evaluate the peroneus muscles after LAS, in recent years, the evaluation of musculoskeletal injuries with ultrasound imaging has evolved into an increasingly standardized technique and has become a reliable tool that can be performed quickly and at a lower cost than other imaging techniques [9]. Ultrasonographic assessment of the morphology of the peroneus muscles has also been commonly used for post-LAS evaluation [10,11,12,13,14]. Morphological changes of the peroneus muscles after LAS have been reported, and the peroneus muscles’ cross-sectional area (CSA) was decreased in individuals with a history of LAS compared to those without a history [11]. In order to properly evaluate musculoskeletal injuries, it is necessary to consider not only muscle CSA, but also the connective tissue around the muscle, such as muscle adipose tissue and fascia, and, as one of the evaluation methods, the effectiveness of echogenicity evaluation using ultrasound has been reported [15]. Previous studies have shown that higher muscle echogenicity indicates a greater proportion of noncontractile tissue in the muscle (i.e., tissue other than muscle fibers, such as muscle adipose tissue) [13]. In addition, one previous study reported that there were differences in the connective tissue status of individuals with a history of LAS compared to healthy individuals, and that these differences may be related to impaired balance, which is a factor in LAS recurrence [16]. Based on these reports, we speculate that using ultrasound to assess the echogenicity of the peroneus muscles in individuals with a history of LAS is important for proper assessment after LAS. Sakai et al. investigated the relationship between the presence or absence of a history of LAS and muscle echogenicity and found that individuals with a history of LAS had a significantly higher peroneus muscle echogenicity than those without a history [14]. From these previous studies, we can predict that the peroneus muscles undergo changes in muscle morphology after LAS, such as a decreased CSA and increased noncontractile tissue, which may be responsible for recurrent LAS.

Conventional measurements of the peroneus muscle morphology via ultrasonography have been performed at only one location, either 25% or 50% proximal to the line connecting the fibular head to the lateral malleolus. However, although the peroneus muscles are long muscles along the long axis of the lateral leg, the distal part has not been measured thus far. The peroneus muscles are mainly divided into the PL and PB, which differ in their origin and insertion. The peroneus muscles are observed at the 50% position, and the morphological changes are examined by measuring the CSA of the PL and PB separately [11]. However, since the PB generally starts at approximately 50% of the fibula length, it may not be possible to accurately observe both muscles on ultrasound images. Since the muscle tissue of the PL ends at the distal 75%, the PB occupies most of the CSA in the distal 75% [17]. It can be inferred that by performing measurements at three locations and adding the distal 75% to the conventional measurement points of the proximal 25% and central 50%, the PL and PB can be evaluated separately, and post-LAS morphological changes in the peroneus muscles can be ascertained more accurately.

When considering morphological changes of the peroneus muscles after LAS, it would be useful to evaluate the peroneus muscle function after LAS, as well as the morphology. Regarding the relationship between morphological and functional muscular characteristics, a study showed a significant positive correlation between the CSA and muscle strength in the quadriceps muscle [18]. Another study showed a significant negative correlation between muscle echogenicity and muscle strength [19], suggesting that there is a relationship between muscle morphology and function. In addition, since it has been reported that a decrease in ankle evertor strength occurs after LAS [6], it is likely that functional changes such as decreased muscle strength affect changes in muscle morphology such as a decreased CSA and an increased muscle echogenicity after LAS. Muscle activity has been reported to be related to muscle strength [20], and since a decrease in muscle activity occurs after LAS [21], corresponding changes in muscle morphology and muscle strength may occur. Evaluating the strength of the ankle joint muscles and the activity of the peroneus muscle, as well as morphological changes after LAS, can provide more detailed information about the morphological changes that occur in the peroneus muscles.

It is important to evaluate the peroneus muscles after LAS, but considering the anatomy of the peroneus muscles, it may be helpful to measure it in more than one place using ultrasonography for a more detailed evaluation. The purpose of this study was to measure the peroneus muscles after LAS at three points, adding distal 75% to the conventional measurement points, in order to obtain a detailed understanding of the post-injury morphology and to propose a new evaluation index of the peroneus muscles for multiple LAS. We hypothesized that legs with a history of LAS would show a decrease in the CSA at all locations, higher muscle echogenicity at all locations, lower muscle activity in both the PL and PB, and lower ankle evertor strength compared to legs without a history of LAS.

## 2. Materials and Methods

### 2.1. Participants

The study protocols complied with the tenets of the Declaration of Helsinki and were approved by the Ethical Committee for Epidemiology of Hiroshima University (approval number: E-2265, assessed on 18 November 2020). All participants provided informed consent for participation in the study.

Figure 1 shows the flowchart of this study. Thirty-two participants were included in this study: 16 with a history of LAS at least twice on the unilateral leg (8 men and 8 women, LAS group) and 16 participants with no history of LAS of the ankle joint (8 men and 8 women, control group). All participants participated in recreational exercises at least once per week. Table 1 shows the profiles of the participants. The inclusion criteria for the LAS group were a history of LAS that required protected weightbearing, immobilization, or limited activity for ≥ 24 h [22]. The exclusion criteria for the LAS group were as follows: (1) a history of bilateral LAS, (2) a history of orthopedic surgery and/or fracture of the lower extremities [23], (3) acute musculoskeletal injuries such as sprains or fractures of joints other than the ankle joints in the lower extremities within the preceding 3 months [23], and (4) experience with rehabilitation for LAS before participating in this study [24]. The exclusion criteria for the control group were previous orthopedic surgery of the lower extremity and acute musculoskeletal injuries such as sprains or fractures of the lower extremity within the preceding 3 months. The participants recruitment period was 1–31 October 2020. Since no previous study had measured the CSA of the peroneus muscles in the distal 75% via ultrasonography, a pre-study sample size was estimated from the CSA at the distal 75% of 3 legs on the LAS side (mean 2.54 cm^2^), 3 legs on the non-LAS side (mean 2.13 cm^2^), and 3 legs of the control participants (mean 1.99 cm^2^) with an alpha level of *p* < 0.05 and a power = 0.8 using G*Power 3.1 (Kiel University, Germany). A minimum of 48 legs was estimated.

### 2.2. Experimental Procedures

All measurements were conducted at Hiroshima University and the measurement period was from 1 November 2020 to 20 December 2020.

#### 2.2.1. Ultrasonographic Imaging of the Peroneus Muscles in the Short-Axis View

A B-mode ultrasound system (ArtUs EXT-1H; Telemed, Vilnius, Lithuania) with an ultrasound probe (5–11 MHz, 60 mm field-of-view; Echoblaster, Telemed, Vilnius, Lithuania) was used to obtain images of the peroneus muscles in the short-axis view. All ultrasound measurements were performed by the same therapist with three years of specialization and experience.

Each participant was placed in a lateral position on the bed, with the ankle joint in the neutral position and a cushion between both lower limbs in a relaxed position. Ultrasound measurements were performed at the proximal 25%, central 50%, and distal 75% of the straight line connecting the fibular head to the lateral malleolus. The gain setting on the ultrasound system, which affects the muscle echogenicity, was set uniformly for all measurements at the distal 75%, which provided clear images [25]. The probe was placed perpendicular to the straight line for measurement and held in a maximally vertical position to minimize the effect of the ultrasound anisotropy. Markings were made on the measurement points to reproduce the points where the probe was applied. After applying sufficient gel to the probe, the probe was placed at the measurement point with minimal force, and the peroneus muscles in the short-axis view were imaged in triplicate at each measurement point (Figure 2).

#### 2.2.2. Measurement of the PL and PB Activity during Isometric Active Ankle Eversion with Maximum Effort Using Surface Electromyography

Surface electromyography (P-EMGplus; Oisaka Electronic Equipment Ltd., Hiroshima, Japan) was used to measure the PL and PB muscle activity during isometric active ankle eversion with maximum effort for 5 s in the maximum plantarflexion position. Measurements were performed at a sampling frequency of 1000 Hz. Surface electrodes (Blue sensor, AmbuA/S; Oisaka Electronic Equipment Ltd., Japan) were used for recording after a thorough skin preparation. Participants were placed in the lateral position on the bed, with the upper leg at 90 degrees hip and knee flexion and the lower leg at 0 degrees to the hip and knee joints [26]. Participants performed isometric active ankle eversion with maximum effort for 5 s in the maximum plantarflexion position. The therapist manually immobilized the participant’s thigh and lower leg to prevent trick movements of the hip and knee joints during the measurement. To normalize the results of muscle activity obtained by ankle eversion in the lateral position, isometric active ankle eversion under maximal effort with manual resistance was performed for 3 s in the sitting position on the bed (Figure 3).

#### 2.2.3. Measurement of Ankle Evertor Strength Using a Handheld Dynamometer

A handheld dynamometer (Mobie; Sakai Medical Co., Ltd., Tokyo, Japan) was used to measure the ankle evertor strength, alongside the measurement of muscle activity. As for the previous measurement, the participant was placed in the same lateral position on the bed [26]. The strap of the handheld dynamometer was placed on the fifth metatarsal head of the foot of the upper leg, and the participant performed an isometric active ankle eversion with maximum effort for 5 s in the maximum plantarflexion position. The reproducibility of this measurement method was confirmed in a previous study [26]. Three measurements were performed bilaterally (Figure 3).

### 2.3. Data Analysis

Image J version 1.52 (National Institutes of Health, Bethesda, MD, USA) was used to calculate the CSA along the fascia on the obtained images. Muscle echogenicity was calculated using the 8-bit-gray-scale, which displayed image brightness in 256 steps from 0 to 255 using Image J. The mean muscle echogenicity of the regions was expressed as a value between 0 (black) and 255 (white) when the CSA was calculated along the fascia on the image. Previous studies have reported that a lower muscle echogenicity indicates a smaller proportion of noncontractile tissue in the muscle that appears darker in ultrasound images (Figure 4) [13]. The average of the three measurements of the CSA and muscle echogenicity was used as the representative value.

Surface electromyography was performed using BIMUTAS-Video (Kissei Comtec Co., Ltd., Nagano Japan), and the recorded waveforms were processed for noise using a 10–500 Hz bandpass filter. Isometric ankle eversion under maximal effort was performed with manual resistance for 3 s in the sitting position on the bed, and the muscle activity during the stable 1 s isometric maximal contraction of the obtained waveform was set at 100%. Next, the results of the muscle activity obtained by ankle eversion in the lateral position were normalized as the percentage of maximum voluntary contraction. The average of three measurements was used as the representative value. The ankle evertor strength was defined as the average of three values obtained from the handheld dynamometer divided by the body weight (N/kg) for standardization to reduce individual differences.

### 2.4. Statistical Analysis

The reproducibility of each measurement was confirmed in the control group using intraclass correlation coefficients (ICC_1,3_). The ICC_1,3_, CSA, and muscle echogenicity were calculated using three images taken at three measurement points. The ICC_1,3_ of muscle activity was calculated from the measurements of three trials of the PL and PB. The ICC_1,3_ of the ankle evertor strength was calculated using measurements from three trials.

The normality of measurement values was confirmed using the Shapiro–Wilk test for the CSA and echogenicity of the peroneus muscles at each measurement point, the PL and PB muscle activities, and the bilateral ankle evertor strengths of the LAS and control groups. When normality was observed, Bartlett’s test was used to evaluate the equality of variance.

Paired *t*-tests were used to compare basic information between the LAS and control groups and to compare the CSA and echogenicity of the peroneus muscles at three measurement points, the PL and PB muscle activities, and the ankle evertor strength within the control group between the left and right sides. After confirming that there was no statistically significant difference between the left and right sides, the mean value of the left and right sides was adopted as the value of the control leg. Each measurement value was compared among the legs from the LAS (LAS side), without LAS (non-LAS side) in the LAS group, and the control leg. One-way analysis of variance was used for comparisons in each group when normality was observed, and Tukey’s post-hoc tests were conducted. The Kruskal–Wallis test was used to compare data with non-normal distributions, and a Steel-Dwass test was conducted as a post-hoc test.

The data were analyzed using SPSS version 27.0 for Windows (SPSS Inc., Chicago, IL, USA). Statistical significance was set at *p* < 0.05.

## 3. Results

Table 2 shows the ICC_1,3_ of the peroneus muscle CSA, echogenicity, PL and PB muscle activities, and ankle evertor strength measurements. The ICC_1,3_ of all measurements was almost perfect, as classified by Landis et al. [27]

Table 3 presents the results of the CSA and echogenicity of the peroneus muscles on the LAS side, non-LAS side, and in the control leg. The CSA of the LAS side was significantly higher than that of the non-LAS side and of the control leg at the distal 75% (*p* = 0.008). The echogenicity was significantly lower in the LAS side at the distal 75% than in the non-LAS side and the control leg (*p* < 0.001). There were no significant differences in either item at the proximal 25% and central 50% in each group.

Table 4 shows the results of the PL and PB muscle activities during ankle eversion and ankle evertor strength on the LAS side, non-LAS side, and in the control leg. In terms of the PL muscle activity, the LAS side showed significantly lower values than the non-LAS side and the control leg (*p* = 0.022). The PB muscle activity was significantly higher on the LAS side than on the non-LAS side and in the control leg (*p* = 0.015), and there was no significant difference in the ankle evertor strength between the groups.

## 4. Discussion

We measured the peroneus muscles at the proximal 25%, central 50%, and distal 75% via ultrasonography among individuals with and without an experience of multiple LASs in one leg only. This is the first study to measure the peroneus muscles at the distal 75% via ultrasonography and to examine the morphological changes after LAS. In addition to the ultrasound measurements, the muscle activity and ankle joint external muscle strength were also measured. The most notable result of this study was that the CSA of the LAS side was significantly higher than that of the non-LAS side and the control leg in the distal 75%. In contrast, there was no significant difference in the CSA of the peroneus muscles between the LAS side, non-LAS side, and in the control leg in the proximal 25% and central 50%.

Considering the anatomical location, the PB occupies most of the CSA in the distal 75% of the measurement points [17]. In other words, the results of this study suggest that the PB may have caused muscle hypertrophy after LAS. This result was a different trend from previous studies that used ultrasound measurements to measure the morphology of the peroneus muscles. Previous reports have measured the peroneus muscles after LAS at either the proximal 25% or the central 50% via ultrasonography. In the proximal 25% and central 50%, where the PL accounts for most of the CSA, the CSA of the peroneus muscles has been considered to decrease [12] or remain unchanged after LAS [10]. The peroneus muscles are long muscles that run along the lateral side of the lower leg, and it is highly predictable that there may be differences in muscle morphology after LAS between the proximal and distal portions. Therefore, it is suggested that it is important to evaluate the peroneus muscles separately for PL and PB by using ultrasound at 75% distal as well as 25% proximal and 50% central, which are the traditional measurement points.

In terms of muscle activity, the results of this study also indicate that muscle activity in the PL decreases and muscle activity in the PB increases, which would be expected to cause morphological changes in the peroneal muscles in the distal 75%. In a previous study, when the triceps surae was excised, the muscle activity of the plantar muscles, whose action is similar to that of the triceps surae, increased compensatively, and the muscles hypertrophied [28]. We speculated that a decrease in the muscle activity of the PL caused a compensatory increase in the muscle activity of the PB, which had the same action as the PL, resulting in muscle hypertrophy. We also predicted that the reason for these results of PL and PB muscle activity in this study could be attributed to the ankle position during the movement of people with a history of LAS. Moisan et al. reported that the inversion angle of the ankle joint increased [29], and an increase in the amount of the lateral load of the foot was observed during movement after LAS [30]. Furthermore, the PL generally increases its activity when the ball of the foot is loaded [31]. In individuals with a history of LAS, the ankle joint position may lead to difficulty in the daily functioning of the PL owing to the increased ankle joint inversion angle and lateral foot loading, resulting in decreased PL muscle activity and increased PB muscle activity. In addition, muscle echogenicity was higher in the distal 75% of the peroneal muscles compared to the non-LAS group. Previous studies have shown that muscles with a large percentage of muscle tissue appear black on ultrasound images [13,32]. It has also been noted that hypertrophied muscles appear more hypoechogenic than healthy subjects on ultrasound images [15]. Since muscle hypertrophy is generally caused by an increase in the CSA of the muscle fibers that make up the muscle due to mechanical stress to the muscle [33], and the percentage of noncontractile tissue decreases as the percentage of contractile tissue increases [34], the results of muscle echogenicity in this study support the result of muscle hypertrophy in PB. Therefore, we think that it is very useful to combine measurements of muscle activity and muscle echogenicity to obtain a detailed understanding of the morphological changes in the peroneus muscles after LAS.

Although ankle evertor strength did not differ according to the presence or absence of a history of LAS, the LAS side showed higher CSA and lower muscle echogenicity at the distal 75% compared with the non-LAS side and control leg in the present study. This may be attributed to the overactivity of the PB after LAS injury compensating for the evertor motion of the ankle joint. A variety of factors affect muscle strength, including muscle morphology, such as the CSA and muscle fiber type, and neurological factors, such as motor units. The ankle evertor strength is frequently evaluated after LAS in clinical practice. However, evaluating the morphological and functional changes in the peroneus muscles after LAS is problematic, as it is difficult to separate the PL and PB based on the results of ankle evertor strength alone. In addition to the ankle evertor strength measurement, a clinical ultrasound measurement is important for detailed assessment of the peroneus muscles after LAS.

Our study had some limitations. First, it did not consider the severity of the LAS, which affects the structural rupture of tissues such as ligaments around the ankle joint, the duration of repair, and the time required to return to sports activity. In the future, we will conduct comparative studies in groups based on the severity of LAS. Second, due to the cross-sectional design of the study, it remains unclear in what sequence changes in the functional and morphological characteristics of the PL and PB occur after LAS. This point needs to be considered to further clarify the relationship between the functional and morphological changes in the peroneus muscles after LAS. In a future study, we will clarify the functional and morphological changes in the peroneus muscles and confirm the relationship between them by following the changes over time after LAS.

## 5. Conclusions

We compared the muscle morphology of the peroneal muscles in the proximal 25%, central 50%, and distal 75% of the lower leg with and without previous LAS, indicating that muscle hypertrophy and overactivity of the PB may have occurred on the LAS side. In order to accurately evaluate the morphological changes of the peroneal muscles after LAS, this study suggested that it is important to measure the peroneal muscles at 75% distal by ultrasonography and to evaluate the peroneal muscles, PL and PB, separately.

## Figures and Tables

**Figure 1 medicina-58-00070-f001:**
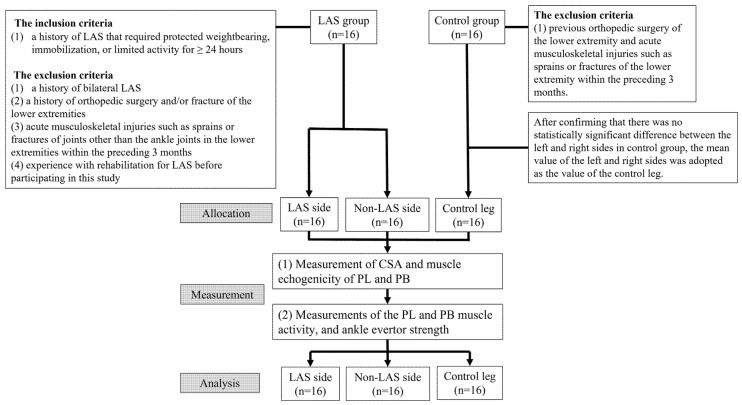
Flowchart of the methodology of this study.

**Figure 2 medicina-58-00070-f002:**
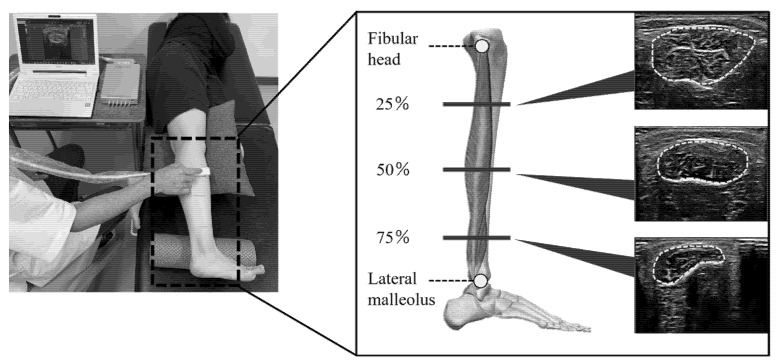
Measurement points of peroneus muscles by ultrasonography. The measurement points were the proximal 25%, central 50%, and distal 75% of the straight line connecting the fibular head and the lateral malleolus.

**Figure 3 medicina-58-00070-f003:**
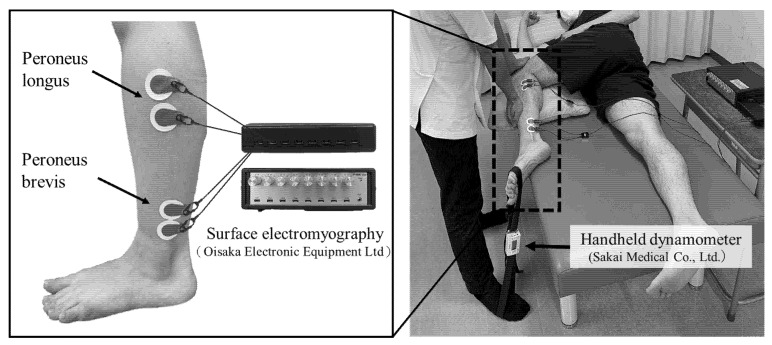
Measurements of the PL and PB muscle activity during ankle eversion by surface electromyography, and ankle evertor strength using a handheld dynamometer. The subject was placed in the lateral position on the bed, with the upper leg at 90° hip and knee flexion and the lower leg at 0° to the hip and knee joints. The belt of the handheld dynamometer was placed on the fifth metatarsal head of the foot of the upper leg, and the subject performed an isometric ankle eversion with maximum effort for 5 s in the maximum plantar flexion position.

**Figure 4 medicina-58-00070-f004:**
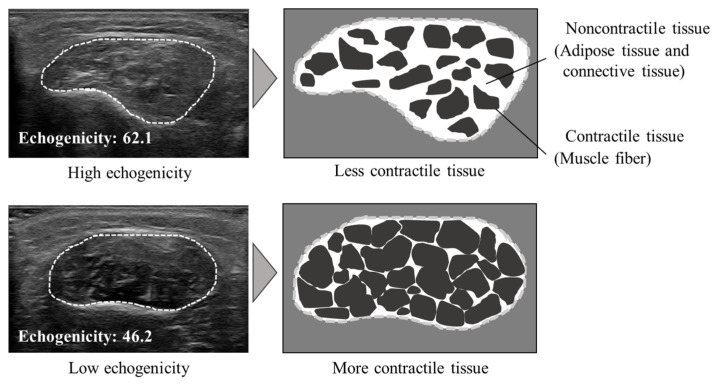
Muscle echogenicity in the ultrasound image. As shown in the upper illustration, the higher the value of muscle echogenicity, the brighter the image appears, indicating more noncontractile tissue in the muscle. As shown in the lower illustration, the lower the value of the muscle echogenicity, the darker it appears in the image, indicating less noncontractile tissue in the muscle.

**Table 1 medicina-58-00070-t001:** Physical characteristics of the subjects.

	LAS Group (*n* = 16)	Control Group (*n* = 16)	*p*-Value
Age (years)	22.3 ± 2.0	22.4 ± 1.5	0.921
Height (cm)	165.1 ± 7.7	166.1 ± 11.2	0.777
Body weight (kg)	58.7 ± 13.2	58.2 ± 12.3	0.906
Body Mass Index (kg/m^2)^	21.3 ± 2.9	20.9 ± 2.7	0.721
The number of ankle sprains (times)	4.3 ± 3.6	0.0 ± 0.0	<0.001

Mean ± SD, LAS: lateral ankle sprain.

**Table 2 medicina-58-00070-t002:** ICC_1,3_ of the peroneus muscles CSA, echogenicity, PL and PB muscle activities, and ankle evertor strength measurements.

	ICC_1,3_	95% CI	SEM	MDC	*p*-Value
**CSA (cm^2^)**					
25%	0.997	0.994–0.999	0.048	0.132	<0.001
50%	0.995	0.990–0.998	0.044	0.121	<0.001
75%	0.995	0.989–0.998	0.025	0.069	<0.001
**Echogenicity (a.u.)**					<0.001
25%	0.977	0.949–0.991	0.735	2.038	<0.001
50%	0.924	0.838–0.970	1.038	2.876	<0.001
75%	0.909	0.807–0.988	1.060	2.939	<0.001
**PL (%MVC)**	0.971	0.936–0.989	3.303	9.155	<0.001
**PB (%MVC)**	0.894	0.778–0.958	3.289	9.107	<0.001
**Ankle evertor strength (N/kg)**	0.908	0.806–0.964	0.093	0.259	<0.001

ICC: intraclass correlation coefficients, ICC (95% CI: confidence interval), CSA: cross-sectional area, a.u.: arbitrary unit, PL: peroneus longus, PB: peroneus brevis, SEM: standard error of measurement, MDC: minimal detectable change.

**Table 3 medicina-58-00070-t003:** The CSA and echogenicity of the peroneus muscles in the LAS and non-LAS sides and in the control leg.

	LAS Side (*n* = 16 Legs)	Non-LAS Side (*n* = 16 Legs)	Control Leg (*n* = 16 Legs)	F-Value	*p*-Value	η^2^
**CSA (cm^2^)**						
25%	4.91 ± 1.00	4.88 ± 0.92	4.89 ± 1.52	0.001	0.999	<0.001
50%	3.61 ± 0.75	3.58 ± 0.72	3.37 ± 1.11	0.332	0.719	0.015
75%	2.54 ± 0.47 *^, †^	2.11 ± 0.40	2.00 ± 0.59	5.321	0.008	0.191
**Echogenicity (a.u.)**						
25%	77.2 ± 8.7	75.4 ± 7.0	79.0 ± 7.6	0.841	0.438	0.036
50%	64.5 ± 8.6	67.0 ± 8.2	69.3 ± 6.8	1.482	0.238	0.062
75%	50.5 ± 9.2 *^, †^	60.9 ± 6.4	61.5 ± 7.3	10.308	<0.001	0.314

Mean ± SD, LAS: lateral ankle sprain, CSA: cross-sectional area, a.u.: arbitrary unit, *: *p* < 0.05 (vs. non-LAS), Tukey test; †: *p* < 0.05 (vs. Control), Tukey test; η^2^ = effect size.

**Table 4 medicina-58-00070-t004:** PL and PB muscle activities during ankle eversion in the LAS and non-LAS sides and in the control leg.

	LAS Side (*n* = 16 Legs)	Non-LAS Side (*n* = 16 Legs)	Control Leg (*n* = 16 Legs)	F-Value	*p*-Value	η^2^
PL (%MVC)	85.0 ± 12.5 *^, †^	94.7 ± 18.1	94.2 ± 10.9	4.166	0.022	0.156
PB (%MVC)	85.7 ± 10.7 *^, †^	75.1 ± 12.9	76.5 ± 8.3	4656	0.015	0.171
Ankle Evertor strength (N/kg)	1.57 ± 0.43	1.92 ± 0.56	1.88 ± 0.48	2.408	0.101	0.097

Mean ± SD, LAS: lateral ankle sprain, PL: peroneus longus, PB: peroneus brevis, *: *p* < 0.05 (vs. non-LAS), Tukey test; †: *p* < 0.05 (vs. Control), Tukey test; η^2^ = effect size.

## Data Availability

The data associated with the paper are not publicly available but are available from the corresponding author on reasonable request.

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
