# Peer review of "Morphological and Functional Characteristics of the Peroneus Muscles in Patients with Lateral Ankle Sprain: An Ultrasound-Based Study"

_medicina, 2022, doi:10.3390/medicina58010070_

Round 1

Reviewer 1 Report

The article is very interesting, well written, pleasant to read and very accessible.

Just very few minor issues to fix:

Line 36 and 84: some words are in bold.

Line 189, Figure 3 legend: the description is inverse. Please correct.

Line 200: "percentage OF maximum..."

Line 294: I would better say: "This study showed that there were changes in muscle activity between the PL and PB"

Line 305: remove the "a" between "higher" and "CSA"

Reviewer 2 Report

The topic is one of importance given the high number of presentations to health services that are related to concerns on  
the prevalence and related factors of lateral ankle sprain in the  population. Also, this is an interesting aim with the investigate assess the morphology of the peroneus muscles after LAS at three locations via ultrasonography to reveal morphological and functional changes and to establish a new evaluation index for multiple LAS to support treatment. I think it would be a more clear study if the following parts were revised and supplemented. These will be discussed below relative to the information of the manuscript.

General Comments:
Overall the manuscript is generally well written and is a topic of interest. However after reading it a number of times I am still left without key take-home points. I believe these points are in the results they just need to be developed.

Specific comments:
Abstract:
1) The authors state they will  assess the morphology of 1he peroneus muscles after LAS at three locations via ultrasonography to reveal morphological and  functional changes and to establish a new evaluation index for multiple LAS to support treatment. This seems like too much of an over simplification of what was done. I do feel that it would be beneficial to explain what specifically you are looking at in relation to lateral ankle sprain (this also applies to the main body of the paper). Is it the development of lateral ankle sprain literature. This needs to be made clearer throughout the paper. (Major Compulsory Revision)

Introduction
2) The first paragraph should have a sentence or two added that better outlines why this study is important related with lateral ankle sprain patients comparison of the Plantar Fascia and Tibialis Anterior in People With and Without Lateral Ankle Sprain https://pubmed.ncbi.nlm.nih.gov/32709515/,  Ultrasonography Comparison of Peroneus Muscle Cross-sectional Area in Subjects With or Without Lateral Ankle Sprains  https://pubmed.ncbi.nlm.nih.gov/27793349/ and electromiography comparison of distal and proximal lower limb muscle activity patterns during external perturbation in subjects with and without functional ankle instability https://pubmed.ncbi.nlm.nih.gov/28843163/

Furthemore, the authors do a poor job on reviewing relevant literatura related with importance with current advances and research in ultrasound imaging to the assessment and management of musculoskeletal disorders. Please revise the research of Romero Morales  et al related with this question https://pubmed.ncbi.nlm.nih.gov/32711897/

3) In the last paragraph, the significance of the proposed word should be included highlighting why your work is important. what is the scientific contribution of this paper? it is not clear how this paper can make a significant contribution to the state of the art. (Major Compulsory Revision).
In addition, author´s hypotheses should be included and to change teh flow chart of the figure one for the method section.

5) This methods section is poor, needs to present a better rationale for the study and the methodology employed. Also, neither appear information related with inclusion and exclusion criteria, dates, protocol. The study design is a experimental research of ramdom sampling method, where the study was conducted in the hospital or in the university center? This research adhere to reporting STROBE guidelines? (Major Compulsory Revision).

6) Where the experiments carried out? In a hospital? In an educational institution? Provide this information.

7) Add figure 1 as a study flow chart for the readers. (Major Compulsory Revision).
8) Include p-values in all the tables (Major Compulsory Revision).
9) The Discussion section is a rehashing of the results. It does not appear that the authors include much interpretation of what the study findings mean for clinical practice or research. (Major Compulsory Revision)

FInally, the conclusión is weak and too long. (Major Compulsory Revision)

Reviewer 3 Report

Title

Title is appropriate because it is completely informative about the contents of the paper.  

Abstract

The abstract respects the rules of the journal. The background and the aim are interesting. In the design is present the type of study.  The clinical Impact is present but need to be better explained, increasing the number of reported references.

Text

The introduction and the discussion of the study does not clearly sum up the background of the study, to add some reference about also the role of connective tissue. The authors provide a rationale for performing the study based on a review of the medical literature. Furthermore, they define well terms used in the remainder of the manuscript. The hypothesis is defined. The methods are clear. The study needs to be better explained in the methodology. The number of references reported about the topic must be increased. For example, the various aspects of the role of the connective tissue in ankle sprains for the diagnosis and management of these disorders can be mentioned in the article introduction and discussion by citing the following articles:” Ultrasound Imaging of Crural Fascia and Epimysial Fascia Thicknesses in Basketball Players with Previous Ankle Sprains Versus Healthy Subjects. Pirri C, Fede C, Stecco A, Guidolin D, Fan C, De Caro R, Stecco C. Diagnostics (Basel). 2021 Jan 26;11(2):177. doi: 10.3390/diagnostics11020177. Stiffness and echogenicity: Development of a stiffness-echogenicity matrix for clinical problem solving. Stecco A, Pirri C, Caro R, Raghavan P. Eur J Transl Myol. 2019 Sep 12;29(3):8476. doi: 10.4081/ejtm.2019.8476.” The results are reported clearly and concisely.

References

They are qualified and updated with the lasted data. The reference list follows the format for the journal.

Tables

They highlight the key points and they are good quality, a different figure would illustrate the findings.

Figures

They highlight the key points and they are good quality, a different figure would illustrate the findings.

Statistical Analysis

It isn’t needed further checking of data by a statistician reviewer.

General comments

The purpose of the study is original but the study needs to be improved in the introduction and discussion. The hypothesis is defined. The methods are clear. The study has been structured and carried out correctly about the methodology. The number of references reported about the topic must be increased. 

Round 2

Reviewer 2 Report

The authors has completed satisfied all the recommendations that I raised and 

stated.

I hope that this publication is available to the public shortly. 

Author Response

Response to Reviewer 2 Comments

I appreciate your thoughtful comments.